

# Sargassum blooms in the Caribbean alter the trophic structure of the sea urchin Diadema antillarum

Nancy Cabanillas-Terán[1], Héctor A. Hernández-Arana[2], Miguel-Ángel Ruiz-Zárate[2], Alejandro Vega-Zepeda[2] and Alberto Sanchez-Gonzalez[3]

[1] Consejo Nacional de Ciencia y Tecnología México- El Colegio de la Frontera Sur, Chetumal, Quintana Roo, México
[2] Departamento de Sistemática y Ecología Acuática, El Colegio de la Frontera Sur, Chetumal, Quintana Roo, México
[3] Centro Interdisciplinario de Ciencias Marinas del Instituto Politécnico Nacional, La Paz, Baja California, México

Corresponding authors
Nancy Cabanillas-Terán,
ncabanillas@ecosur.mx
Héctor A. Hernández-Arana,
hhernand@ecosur.mx

## ABSTRACT

The arrival of large masses of drifting *Sargassum* since 2011 has caused changes in the natural dynamics of Caribbean coastal ecosystems. In the summer of 2015, unprecedented and massive mats of *S. fluitans* and *S. natans* have been observed throughout the Mexican Caribbean including exceptional accumulations ashore. This study uses stable isotopes to assess the impact of *Sargassum* blooms on the trophic dynamics of the *Diadema antillarum* sea urchin, a keystone herbivore on many Caribbean reefs. Bayesian models were used to estimate the variations in the relative proportions of carbon and nitrogen of assimilated algal resources. At three lagoon reef sites, the niche breadth of *D. antillarum* was analysed and compared under massive influx of drifting *Sargassum* spp. vs. no influx of *Sargassum* blooms. The effects of the leachates generated by the decomposition of *Sargassum* led to hypoxic conditions on these reefs and reduced the taxonomic diversity of macroalgal food sources available to *D. antillarum*. Our trophic data support the hypothesis that processes of assimilation of carbon and nitrogen were modified under *Sargassum* effect. Isotopic signatures of macroalgae associated with the reef sites exhibited significantly lower values of $\delta^{15}$N altering the natural herbivory of *D. antillarum*. The Stable Isotopes Analysis in R (SIAR) indicated that, under the influence of *Sargassum* blooms, certain algal resources (*Dictyota*, *Halimeda* and *Udotea*) were more assimilated due to a reduction in available algal resources. Despite being an abundant available resource, pelagic *Sargassum* was a negligible contributor to sea urchin diet. The Stable Isotope Bayesian Ellipses in R (SIBER) analysis displayed differences between sites, and suggests a reduction in trophic niche breadth, particularly in a protected reef lagoon. Our findings reveal that *Sargassum* blooms caused changes in trophic characteristics of *D. antillarum* with a negative impact by hypoxic conditions. These dynamics, coupled with the increase in organic matter in an oligotrophic system could lead to reduce coral reef ecosystem function.

## INTRODUCTION

The arrival of massive amounts of pelagic *Sargassum* spp. has caused changes in the natural benthic dynamics of Caribbean coastal ecosystems for the last nine years (*Gower, Young & King, 2013*; *Schell, Goodwin & Siuda, 2015*). Pelagic *Sargassum* is a complex of two species, namely *S. fluitans* and *S. natans* (*Oyesiku & Egunyomi, 2014*). Since 2011, extensive masses of *Sargassum* appeared in unusual ways in oceanic waters off northern Brazil (*De Széchy et al., 2012*; *Sissini et al., 2017*), along the West Indies and Caribbean coasts (*Gower, Young & King, 2013*) from Trinidad to the Dominican Republic (*Rodríguez-Martínez, van Tussenbroek & Jordán-Dahlgren, 2016*; *van Tussenbroek et al., 2017*), and along the west African coast from Sierra Leone to Ghana (*Smetacek & Zingone, 2013*). *Wang et al. (2019)* recorded that for June 2018, wet biomass reached more than 20 million tons in the Caribbean Sea and Central Atlantic Ocean.

The Mexican Caribbean shores faced atypical massive mats of pelagic *Sargassum* in the summer of 2015 (*van Tussenbroek et al., 2017*; *Cuevas, Uribe-Martínez & Liceaga-Correa, 2018*; *Arellano-Verdejo, Lazcano-Hernandez & Cabanillas-Terán, 2019*). There was a subsequent decrease during 2016 and 2017, but for most of 2018 and thus far in 2019 influx has increased again. Several studies revealed that these massive mats of *Sargassum* have a new possible distribution source different from the historic North Atlantic Recirculation Region (NARR) known as "The Sargasso Sea" (*Schell, Goodwin & Siuda, 2015*). Instead, the most probable origin of the massive influx on the Caribbean shores is the North Equatorial Recirculation Region (NERR) (*Johnson et al., 2013*; *Schell, Goodwin & Siuda, 2015*). High oceanic temperatures and nutrient inputs (*Franks, Johnson & Ko, 2016*; *Wang et al., 2018*), among other oceanographic coupled patterns such as changes of surface currents, are the most probable causes of this new region of *Sargassum* flourishment (*Johnson et al., 2013*; *Gower, Young & King, 2013*; *Sissini et al., 2017*). A recent study by *Wang et al. (2019)* revealed that increases of pelagic *Sargassum* are driven by upwelling off West Africa during the boreal winter and by Amazon River discharge during the spring and summer. The authors state that recurrent blooms in the Caribbean Sea and tropical Atlantic are likely, highlighting the importance for understanding their effects on existing ecosystems for future planning.

Changes in habitat structure can directly influence trophic dynamics (*Hunter & Price, 1992*; *Sweatman, Layman & Fourqurean, 2017*) and have been shown to cause synergistic effects on coral reefs (*Smetacek & Zingone, 2013*). For example, harmful macroalgae blooms have been recognized as drivers of degradation in coral reef habitats (*Lapointe et al., 2005*). This has effects on the diversity of reef biota (*Bauman et al., 2010*; *Louime, Fortune & Gervais, 2017*) like variations in the sea urchin populations (*Lapointe et al., 2010*). The carbon and nitrogen flow by macroalgae blooms likely has adverse effects at different scales. Such disturbances from *Sargassum*, coupled with pre-existing threats on coral reefs, add to the drivers of Anthropocene reef degradation (*Alvarez-Filip et al., 2011*; *Cramer et al., 2012*).

The massive decomposition of *Sargassum* has negative impacts not only on tourism and local fisheries, but on nearshore ecosystems (*Solarin et al., 2014*; *Louime, Fortune &*

*Gervais, 2017*). However, few studies assess the trophic impact of *Sargassum* blooms on benthic communities. Pelagic *Sargassum* and their attached epiphytic algae can contribute new organic matter to these communities (*Rooker, Turner & Holt, 2006*; *Wang et al., 2018*). Therefore, we consider whether or not these new sources of nitrogen and carbon act in a detrimental manner on the trophic chain of benthic communities. The beaching and decomposing of massive *Sargassum* mats produce hypoxia in near-shore coral reef communities (*Rodríguez-Martínez et al., 2019*). This effect coupled with high hydrogen sulfide and ammonium concentrations have been shown to cause faunal mortality in the Mexican Caribbean (*Rodríguez-Martínez et al., 2019*). As a consequence, the coastal environment becomes even more sensitive to degradation agents. To assess these issues, we included measurements of dissolved oxygen in our study.

Evaluating consumers and resources through a trophic approach by tracking the relationships between consumers and prey provides relevant information on the trophic structure and dynamics of a benthic community (*Minagawa & Wada, 1984*; *Vanderklift, Kendrick & Smit, 2006*; *Behmer & Joern, 2008*). Stable isotopes of carbon ($\delta^{13}$C) and nitrogen ($\delta^{15}$N) have been used in marine ecosystems to determine the feeding habits of species (*Peterson & Fry, 1987*), nutrient migrations within food webs, trophic position of organisms and their contribution at all trophic levels (*Vander Zanden & Rasmussen, 1996*). It is also possible to trace the origin and transformation of the ingested organic matter and to detect changes in the trophic positions of organisms that coexist in the same habitat (*Hobson, 1999*; *Vanderklift, Kendrick & Smit, 2006*; *Rodríguez-Barreras et al., 2016*).

Stable carbon and nitrogen isotope ratios provide time-integrated information regarding feeding relationships and energy flow through food webs (*DeNiro & Epstein, 1981*; *Peterson & Fry, 1987*; *Vander Zanden & Rasmussen, 2001*). Moreover, stable isotopes can be used to study the trophic niche breadth of a species (*Bearhop et al., 2004*; *Parnell et al., 2010*; *Phillips et al., 2014*). This is directly influenced by consumers and resource input, providing a quantitative assessment of trophic conditions (*Newsome et al., 2007*; *Boecklen et al., 2011*). Stable isotope analyses are useful for assessing the health of ecosystems because it is possible to associate the consumers trophodynamics and niche breadth with habitat disturbances (*Layman et al., 2007b*; *Hamaoka et al., 2010*). It is also possible to detect changes in the trophic spectrum from anthropogenic impacts or unusual conditions that cause shifts in ecosystems (*Wing et al., 2008*; *Prado, Alcoverro & Romero, 2010*; *Tomas, Box & Terrados, 2011*). In light of the massive arrival of pelagic macroalgae, sea urchin herbivory is a good model to understand variability in the benthic trophic chain, as sea urchins are considered generalist consumers with a plastic feeding habit (*Lawrence, 1975*; *Vanderklift, Kendrick & Smit, 2006*). Echinoids have the capability to modify the community structure through foraging behaviour (*Carpenter, 1986*; *Hay & Fenical, 1988*; *Sala et al., 1998*; *Eklöf et al., 2008*). Thus, the relative position of $\delta^{13}$C vs. $\delta^{15}$N echinoids displayed in a bi-plot can give insights about organism responses to niche shifts, diet variability and habitat modification (*Layman et al., 2007a*; *Layman et al., 2007b*; *Layman et al., 2012*; *Sweatman, Layman & Fourqurean, 2017*).

The effect of *Sargassum* and their leachates on the diet of *D. antillarum* can improve our understanding on the impact on trophic ecology of one of the most important sea

urchins of the Mexican Caribbean. The main reason to focus this study on *D. antillarum* is that this species is and was the major shallow-hard-bottom grazer in our study sites (*Jorgensen, Espinoza-Ávalos & Bahena-Basave, 2008*; *Jordán-Garza et al., 2008*). One of the most dramatic events in the Caribbean resulted from the pathogen-driven reduction in the populations of *D. antillarum* (*Lessios et al., 1984*) with detrimental ecological consequences like coral-algal phase-shifts. The southern part of Quintana Roo is not an exception encompassing with the effects of the abrupt coastal development and watershed pollution as key drivers along the Costa Maya (*Arias-González et al., 2017*).

The overarching aim of this study was to determine variations in the relative proportions of carbon and nitrogen of assimilated algal resources and the niche breadth of *D. antillarum* under massive influx of drifting *Sargassum* spp. vs. no influx of *Sargassum* at back reefs. We also aimed to determine whether pelagic *Sargassum* was a substantial source of energy for *D. antillarum*. To do this, we compared $\delta^{15}$N and $\delta^{13}$C values of *D. antillarum* with and without influx of *Sargassum* to track changes in this species trophic ecology (diet, trophic position and niche breadth). Ultimately, we tested the hypothesis that an influx in *Sargassum* in coastal ecosystem creates a significant change in the available algal sources and a shift in the trophic structure.

## MATERIAL & METHODS

### Study sites

We determined the stable isotopes of carbon and nitrogen for *D. antillarum* at three reef lagoons (Mahahual, Xahuayxol, and Xcalak) with different distances from the beach to the reef crest (Fig. 1). The main strategy implemented by local authorities at some beaches with the massive arrival of macroalgae included the removal and disposal of *Sargassum* in the highest part of the beach or in places determined *ex profeso*. This contributed to a continuous accumulation of *Sargassum* masses on the beach. However, the *Sargassum* removal was not quantified and the information regarding removal included here is only preliminary.

Mahahual (18°42′16.96″N 87°42.619′W) is located in the northern part of the Mesoamerican Barrier Reef System (MBRS) in the state of Quintana Roo. Mahahual is a former fishing village but during the last two decades has undergone reef degradation due to anthropogenic impact (*Martínez-Rendis et al., 2016*). It has a narrow reef lagoon (230–450 m). *Sargassum* management in this locality was active through removing it from the beach and *ex situ* disposition.

Xahuayxol (18°30′21.78″N; 87°45′24.84″W) located south of Mahahual, has a larger reef lagoon measuring 300 to 500 m from the beach to the reef crest. *Sargassum* was not removed from the beach in any systematic way and remained accumulated on the shore. This reef is the northern limit of the marine protected area Parque Nacional Arrecifes de Xcalak (PNAX) and human activities are less salient than in Mahahual (*Schmitter-Soto et al., 2018*).

Xcalak (18°14′7.68″N; 87°50′1.46″W), at the southern limit of the Mexican Caribbean, is part of PNAX since 2000. It is also part of the MBRS (*Hoffman, 2009*). It has a wide reef
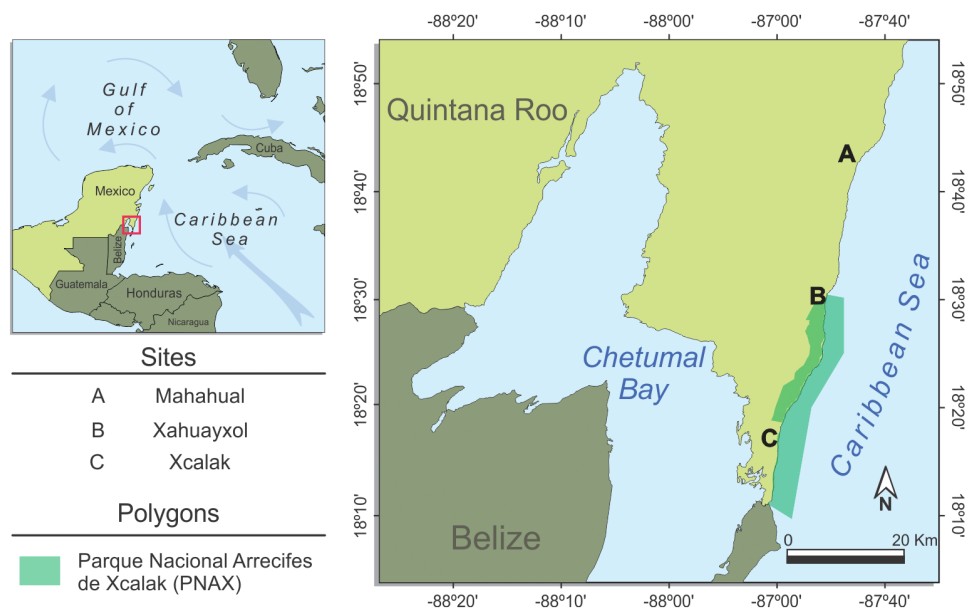

**Figure 1** **Study sites.** Study area and sampling localities at the south coast of Quintana Roo: Mahahual (A), Xahuayxol (B) and Xcalak (C). The green polygon represents the marine protected area Parque Nacional Arrecifes de Xcalak (PNAX). Figure credit: Alejandro A. Aragón-Moreno.

lagoon (950–1,200 m), and *Sargassum* was accumulated along the shore in large amounts. There was active but less intense *Sargassum* management in place at Xcalak, where final disposal was *in situ* on the highest part of beach.

At all sampled sites, the dominant forcing mechanism was reef lagoon circulation from wave action (*Mariño-Tapia et al., 2010*). In our study area, during the period from June to August has the wave orbital velocity over the threshold of motion (*Maldonado-Sánchez et al., 2019*), indicating active circulation in the reef lagoons.

## Collecting and processing data

This study covers two periods: Under *Sargassum* effect (USE) during the months of July–August 2015 and without *Sargassum* effect (WSE) in July–August 2016. USE sampling for stable isotope analysis included drifting *Sargassum* (mixture of S. *fluitans* and S. *natans*), turf associated pelagic *Sargassum*, benthic macroalgae, local turf and 19 individuals of *D. antillarum*. WSE sampling included benthic macroalgae, local turf and 15 individuals of *D. antillarum* (see sampling details ST1). Samples sizes were based on previous studies to obtain sufficient data for statistical analysis (*Rodríguez, 2003*; *Tomas et al., 2006*; *Wing et al., 2008*; *Rodríguez-Barreras et al., 2016*). The sampling sites were at coastal lagoons in the back reef zone (section c, Fig. 2), zone with no visible presence of *Sargassum* leachates (*van Tussenbroek et al., 2017*) and where *D. antillarum* is distributed (*Steneck & Lang, 2003*; *Jorgensen, Espinoza-Ávalos & Bahena-Basave, 2008*; *Jordán-Garza et al., 2008*; *Maldonado-Sánchez, 2018*).

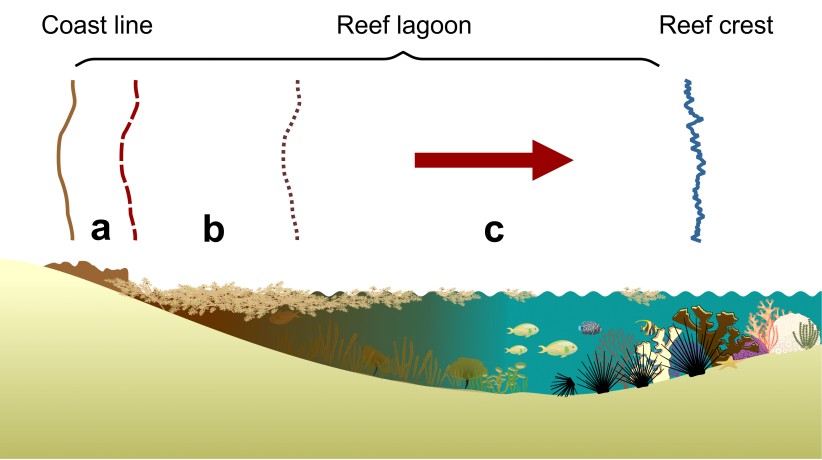

**Figure 2** **Lagoon reef-scape showing the sections with *Sargassum* blooms.** Lagoon reef-scape showing the sections: a: decomposing *Sargassum* spp., Section b, leachates (dark brown water) and section c, back reef, areas without visible leachates. Based on *van Tussenbroek et al. (2017)*.

## Under *Sargassum* effect (USE) measurements

USE included measurements of dissolved oxygen (mg l$^{-1}$) recorded with a calibrated Multi-parameter water quality checker HORIBA 50 at Mahahual, Xahuayxol and Xcalak. Measurements of dissolved oxygen were made at points distributed in three sections from areas with decomposing *Sargassum* (section a), leachates (section b -dark brown water-) and reef lagoon areas without *Sargassum* leachates (section c) (Fig. 2).

Pelagic *Sargassum* spp.*,* turf (benthic turf and the associated turf to pelagic *Sargassum*) and macroalgae samples were collected in coral reef patches of section c (back reef zone) for each sampling site.

## Under and without *Sargassum* effect (USE and WSE) measurements

We collected algal samples to obtain biomass, and for stable isotope analysis using nine quadrats (50 × 50 cm) per site. Pelagic *Sargassum* biomass was calculated based on sunken thalli and overlaid on reef substrates inside the quadrats. The quadrats were located randomly within the sea urchin habitat (radius of 15 m from collected echinoids). The substrate inside each quadrat was scrapped, carefully removed, collected in bags, and frozen for later analysis.

Macroalgae were identified according to *Littler & Littler (2000)*. Analyses were performed to genus level. For biomass estimates samples were dried for 48 h in an oven at 60 °C. Samples were weighed with a digital balance (standard error = 0.0001 g). To determine *D. antillarum* differential algae assimilation considering USE and WSE, algae samples were pooled per site. The sampled echinoids and algal species for this study are not threatened. The collection permit was obtained from the Comisión Nacional de Acuacultura y Pesca (CONAPESCA, PPF/DGOPA-002/17).

The collected individuals of *D. antillarum* were at the same depth range (1.5–2.5 m) and only individuals greater than 5.0 cm in test diameter were collected to avoid any ontogenic effect. Samples were frozen shortly after collection and processed later at the

laboratory. The muscles of Aristotle's lanterns were carefully removed and washed from the stomach contents to estimate algal assimilation by *D. antillarum* because this tissue offers a time-integrated measure of carbon and nitrogen assimilated sources (*Polunin et al., 2001*; *Ben-David & Schell, 2001*; *Phillips & Koch, 2002*).

Macroalgae and local turf, pelagic *Sargassum* species (*S. fluitans* and *S. natans*), turf associated to pelagic *Sargassum*, and echinoids muscle samples were rinsed with filtered water, dried at 50 °C during 48 h, grounded to a fine powder and placed in glass vial for isotope analyses. To remove carbonates from some algal species (eg., *Halimeda* spp. *Penicillus* spp., etc.), the samples were washed with diluted HCl at 1 N prior to drying to avoid disturbance in the mass spectrometer reading.

A subsample of each algae and muscle (1mg) was taken to evaluate the $^{13}C/^{12}C$ and $^{15}N/^{14}N$ ratios using a Delta V Plus Mass Spectrometer. Catalyzers silvered cobaltous/cobaltic oxide and chromium oxide were used. Carbon and nitrogen samples were analysed in a dual isotope mode at the Centro Interdisciplinario de Ciencias Marinas from Instituto Politécnico Nacional. Isotope samples were loaded into tin-capsules and placed in a 50-position automated Zero Blank sample carousel on a COSTECH 4020 elemental analyzer. The carbon and nitrogen isotopic results were expressed in standard delta notation relative to Vienna Pee Dee Belemnite (VPDB) and to atmospheric air.

$$\delta^{13}C = \left[ \left( \frac{\left(\frac{^{13}C}{^{12}C}\right) \text{Sample}}{\left(\frac{^{13}C}{^{12}C}\right) \text{Standard}} \right) - 1 \right] \times 1000\, (\text{\textperthousand})$$

and

$$\delta^{15}N = \left[ \left( \frac{\left(\frac{^{15}N}{^{14}N}\right) \text{Sample}}{\left(\frac{^{15}N}{^{14}N}\right) \text{Standard}} \right) - 1 \right] \times 1000\, (\text{\textperthousand}).$$

The standard deviations of $\delta^{13}C$ and $\delta^{15}N$ replicate analyses were estimated; the precision values were 0.2‰ for carbon and nitrogen isotope measurements. In addition, we calculated the trophic level (TL) according to *Hobson & Welch (1992)* for every individual of *D. antillarum* in each site, expressed as:

$$TL = \frac{1 + (Nm - Nb)}{TEF}.$$

Where Nm is the mean $\delta^{15}N$ ratio of each sea urchin, Nb is average basis $\delta^{15}N$ value of the algal community, and TEF is the given value for the trophic enrichment factor (TEF). We assumed a TEF of 2.4 following *Moore & Semmens (2008)*.

## Data analysis

Dissolved oxygen data were summarized to obtain average values (± standard error) by section (sections a, b, c in Fig. 2) and reef lagoons (Mahahual, Xahuayxol, and Xcalak). We evaluated differences among sections and at the reef lagoons (sections a, b, c, in Fig. 2). We plotted raw data of dissolved oxygen as a function of distance to coast to visualize the low to high values gradient related to that distances in every reef lagoon.

The relative contribution of algae to the diet of the sea urchins *D. antillarum* was estimated with a Bayesian isotopic mixing model (SIAR *Parnell & Jackson, 2013*), which included the isotopic signatures, fractionation and variability to estimate the probability distribution of the contribution of the food source to a mixture. This procedure supplied accurate information about the contribution of algal species to the sea urchin tissues, as it provided the proportion for every source and recognized the main sources as important components of the diet (*Peterson, 1999*; *Fry, 2006*; *Wing et al., 2008*) at three different sites, and under and without *Sargassum* effect. To run the model, the isotopic discrimination factor values used were 2.4 ± 1.6‰ (mean ± SD) for $\delta^{15}$N, and 0.4 ± 1.3‰ (mean ± SD) for $\delta^{13}$C (*Minagawa & Wada, 1984*; *Fry & Sherr, 1989*; *Moore & Semmens, 2008*; *Cabanillas-Terán et al., 2016*).

The following algal taxa/groups were considered for the mixing models analyses: *Caulerpa, Codium, Dictyota, Halimeda, Laurencia, Lobophora, Padina, Penicillus, Sargassum polyceratum, Stypopodium,* turf, and *Udotea.* The sources for the model were selected following the theoretical geometric assumptions of the mixing model according to *Phillips et al. (2014)* and *Rodríguez-Barreras et al. (2015)* to ensure reliable resources. Samples of *D. antillarum* did not require lipid extraction since C:N ratios of Aristotle lantern's muscle were lower than 3.5 (*Post et al., 2007*).

We performed a comparison USE and WSE between the niche width and overlap for *D. antillarum* by using Stable Isotope Bayesian Ellipses in R (SIBER) (*Jackson et al., 2011*) from the SIAR package (*Parnell & Jackson, 2013*). This procedure performs metrics based on ellipses and provides the standard ellipse corrected area (SEAc) used as the trophic niche breadth and the overlap between ellipses, presuming that values close to 1 exhibit a higher trophic overlap. Models were run with 200,000 iterations and a burn in of 50,000.

Homogeneity and normality of variance were tested by performing a Kolmogorov–Smirnov and a Cochran's test (*Zar, 1999*). Nitrogen data followed the premises of parametric analysis, but the carbon, dissolved oxygen and biomass data required a power transformation for reaching normality and homogeneity of variance (*Box & Cox, 1964*). We ran two-way ANOVA to evaluate dissolved oxygen data differences among sections in the reef lagoons and we performed a post hoc comparison using Tukey-HSD test. The functions aov and glm from the Gaussian family were used to test the differences in isotopic ratios of carbon and nitrogen values to compare the effect (WSE and USE) between sites and their interaction. Statistics were performed with $\alpha < 0.05$ (R Core Team, 1.0.153, 2017).

## RESULTS

The dissolved oxygen values USE indicated that the effects of the leachates generated by the decomposition process, together with the organic material carried in their vegetal structures, reduced the values of dissolved oxygen in the reef lagoon water. The decomposing *Sargassum* area (section a, Fig. 2) showed an average range from 1.01 (S.E. ± 0.30) mg l$^{-1}$at Xcalak to 1.88 (S.E. ± 0.37) mg l$^{-1}$ at Mahahual. The leachates area (section b, Fig. 2) showed an average range from 2.42 (S.E. ± 0.32) mg l$^{-1}$ at Xahuayxol to 3.66 (S.E. ± 0.42)

mg l$^{-1}$at Mahahual. The back reef area (section c, Fig. 2) showed an average range from 4.1 (S.E. $\pm$ 0.34) mg l$^{-1}$at Mahahual to 4.8 (S.E. $\pm$ 0.22) mg l$^{-1}$at Xcalak. The two-way ANOVA indicated significant differences between reef lagoons ($p < 0.05$) and sections ($p < 0.01$); Mahahual was significantly different to Xcalak, but Mahahual and Xcalak were not significantly different to Xahuayxol (Post-hoc HSD of Tukey test, 95% confidence). The three sections at the three reefs were significantly different, except the sections b and c of Mahahual (Post-hoc HSD of Tukey test, 95% confidence). Therefore dissolved oxygen data showed a gradient significantly different between sections. The overall values of dissolved oxygen displayed the lowest concentrations for section a, near the shoreline and higher values beyond the back reef section c (Fig. 3).

## Biomass, $\delta^{15}$N and $\delta^{13}$C of macroalgae

Biomass data for benthic taxa displayed no significant differences between USE and WSE, but significant differences were found among localities (ANOVA, $df = 2$, $F = 8.24$, $p < 0.0001$). Mahahual had the highest mean benthic biomass values (55.2 dry weight m$^{-2}$) followed by Xahuayxol with (38.8 dry weight m$^{-2}$) and Xcalak (16 dry weight m$^{-2}$ $\pm$). WSE biomass average values for local benthic algae ranged from 3.01 dry weight m$^{-2}$ $\pm$0.95 (*Codium* spp. at Xcalak) to 133.50 dry weight m$^{-2}$ $\pm$30.29 (*Halimeda* spp. at Mahahual). USE values ranged from 7.75 dry weight m$^{-2}$ $\pm$5.4 (*Caulerpa* at Xcalak) to 145.99 dry weight m$^{-2}$ $\pm$ 36.21 (*Halimeda* spp. at Mahahual, Table 1). Genus-level biomass of pelagic taxa showed no significant differences per site neither at genus level, however *Sargassum fluitans* displayed the highest biomass values.

Under and without *Sargassum* effect values revealed significant differences in overall benthic algae values of $\delta^{15}$N (ANOVA, $df = 1$, $F = 20.27$, $p < 0.0001$). Specifically under *Sargassum* blooms most of the algae exhibited isotopic signatures with significantly depleted $\delta^{15}$N like *Dictyota* and turf across the lagoon reef sites (Table 2). The overall macroalgal $\delta^{15}$N under *Sargassum* fluctuated from 0.023 to 2.08‰. At Xcalak *Caulerpa* displayed the highest mean values of nitrogen with 2.02 $\pm$ 0.08‰. Local Turf USE displayed negative values and overall turf values fluctuated from −0.97‰ to 0.42‰. Xahuayxol displayed the most negative $\delta^{15}$N mean value of local turf (−0.51 $\pm$ 0.02‰). Without *Sargassum* effect the mean algal genus $\delta^{15}$N fluctuated from 0.06 $\pm$ 0.08 with *Penicillus* at Xcalak, and Xahuayxol displayed the highest mean value of $\delta^{15}$N with *Caulerpa* (5.68 $\pm$ 0.01‰) (Table 2).

As for $\delta^{13}$C USE ratios fluctuated from −21.98 to −9.23‰ and WSE from −20.90 to −5.65‰. Considering only the algae presented in both sampling periods (WSE and USE) there was no significant difference in $\delta^{13}$C among sites (ANOVA, $df = 2$, $F = 0.55$, $p > 0.05$) neither was significant difference analysing the effect (ANOVA, $df = 1$, $F = 1.14$, $p > 0.05$) and their interaction (ANOVA, $df = 2$, $F = 0.86$, $p > 0.05$).

Overall USE pelagic *Sargassum* $\delta^{13}$C values fluctuated from −17.95‰ to −15.24‰. *S. natans* exhibited the most negative mean values of $\delta^{13}$C (−17.44 $\pm$ 0.71‰) at Mahahual (Table 2). There was no difference in $\delta^{13}$C among sites (ANOVA, $df = 2$, $F = 0.05$, $p > 0.05$) but there were significant differences $\delta^{13}$C between species (ANOVA, $df = 2$, $F = 7.57$, $p = 0.01$). *Sargassum*'s associated turf $\delta^{13}$C values fluctuated from −18.65‰

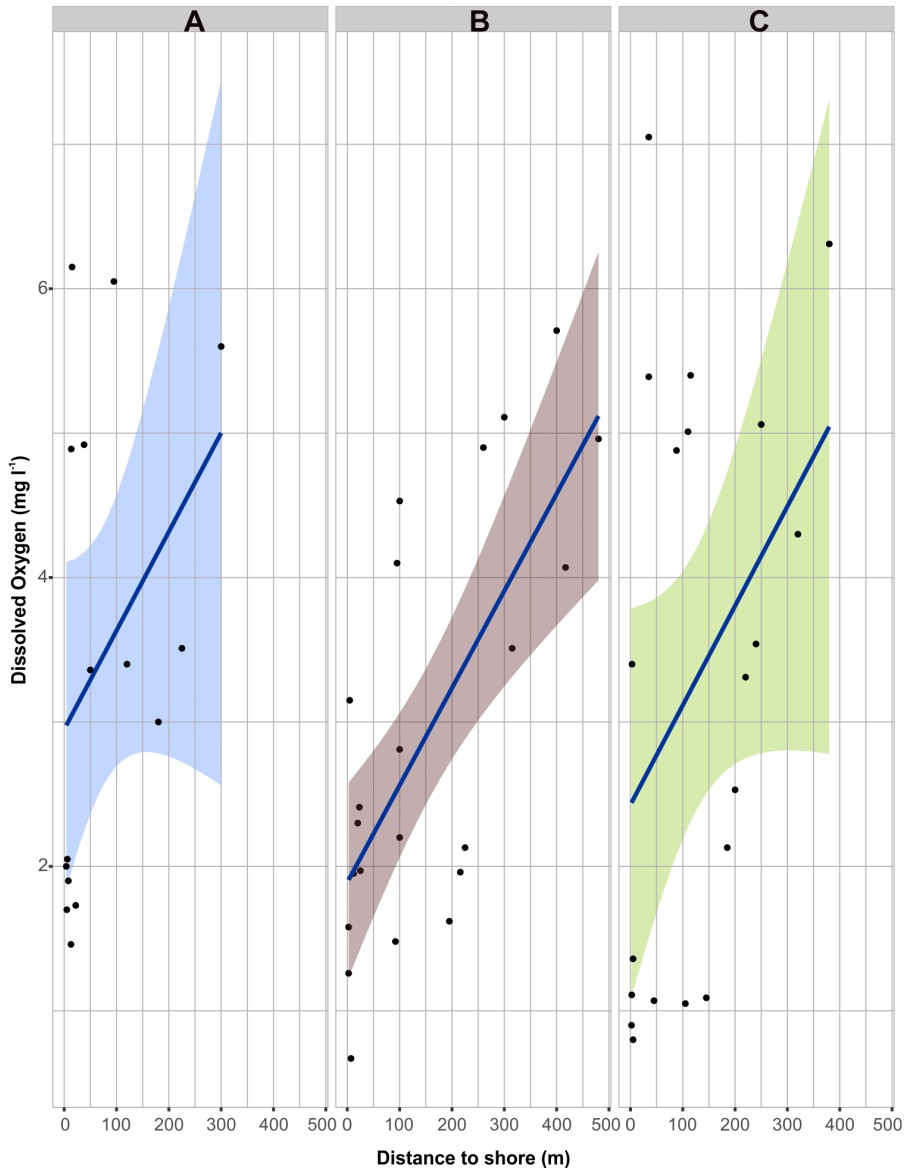

**Figure 3** **Dissolved Oxygen values under Sargassum effect (USE).** Dissolved oxygen (mgl$^{-1}$) values along the distance to shoreline at (A) Mahahual (blue), (B) Xahuayxol (purple) and (C) Xcalak (green) considering the sections depicted in Fig. 2: dissolved oxygen < 2 mgl$^{-1}$: decomposing *Sargassum* spp; dissolved oxygen between 2–4 mgl$^{-1}$: leachates (dark brown water) and dissolved oxygen > 4 mgl$^{-1}$: back reef, areas without visible leachates.

to −15.37‰. The most negative δ$^{13}$C mean value was displayed at Mahahual (−18.3 ± 0.5‰) for *Sargassum*'s associated turf.

Overall pelagic *Sargassum* δ$^{15}$N values ranged from −2.87‰ to −0.30‰. The less negative mean value was exhibited at Mahahual (−0.53 ± 0.26‰) for *S. fluitans.* There was no significant difference for δ$^{15}$N among sites (ANOVA, $df = 2$, $F = 3.90$, $p = 0.05$), but there was a remarkable trend to depleted δ$^{15}$N at Xcalak where *S. fluitans* displayed

Cabanillas-Terán et al. (2019), *PeerJ*, DOI 10.7717/peerj.7589

**Table 1  Algal biomass values.** Mean ± standard deviation values of algal biomass (grams dry weight m$^{-2}$) at Mahahual, Xahuayxol and Xcalak. Genus considered for the mixing models analysis. Data below the grey line belongs to pelagic taxa.

| Mahahual | | | Xahuayxol | | | Xcalak | | |
|---|---|---|---|---|---|---|---|---|
| **Genus** | **WSE** | **USE** | **Genus** | **WSE** | **USE** | **Genus** | **WSE** | **USE** |
| *Caulerpa* | 39.49 ± 20.79 | 19.82 ± 6.48 | *Caulerpa* | 5.38 ± 0.93 | | *Caulerpa* | 7.97 ± 3.51 | 7.75 ± 5.4 |
| *Dictyota* | 19.92 ± 11.69 | 20.40 ± 5.41 | *Dictyota* | 6.61 ± 2.49 | 20.36 ± 5.96 | *Codium* | 3.01 ± 0.95 | |
| *Halimeda* | 133.50 ± 30.29 | 145.99 ± 36.21 | *Halimeda* | 118.07 ± 29.43 | 89.18 ± 9.998 | *Dictyota* | 21.99 ± 5.99 | 17.76 ± 2.34 |
| *Laurencia* | 14.73 ± 22.15 | | *Laurencia* | 8.49 ± 4.10 | | *Lobophora* | 26.96 ± 4.30 | |
| *Stypopodium* | 95.41 ± 66.10 | | *Lobophora* | | 19.933 ± 11.50 | *Padina* | 12.62 ± 4.30 | |
| Turf | 24.69 ± 9.17 | 19.042 ± 6.045 | *Penicillus* | 12.88 ± 3.94 | | *Penicillus* | 27.49 ± 3.51 | 26.23 ± 2.45 |
| *Udotea* | 59.79 ± 45.74 | | *Sargassum* | 14.26 ± 4.42 | | *Sargassum* | 15.01 ± 4.30 | |
| | | | *Stypopodium* | 10.06 ± 12.13 | | Turf | 12.00 ± 3.51 | 11.40 ± 4.21 |
| | | | Turf | 5.886 ± 2.83 | 14.26 ± 7.84 | | | |
| | | | *Udotea* | 39.13 ± 14.76 | 34.02 ± 16.54 | | | |
| *S. fluitans* | | 12.39 ± 8.33 | *S. fluitans* | | 11.86 ± 2.75 | *S. fluitans* | | 13.00 ± 6.99 |
| *S. natans* | | 4.92 ± 3.14 | *S. natans* | | 7.07 ± 3.26 | *S. natans* | | 10.03 ± 7.94 |
| *Sargassum*'s associated turf | | 3.10 ± 1.21 | *Sargassum*'s associated turf | | 3.23 ± 1.28 | *Sargassum*'s associated turf | | 1.98 ± 1.29 |

Cabanillas-Terán et al. (2019), *PeerJ*, DOI 10.7717/peerj.7589
**Table 2** Mean ± standard deviation values of δ¹³C and δ¹⁵N of algal genus considered in the mixing model analysis taken from Mahahual, Xahuayxol and Xcalak, the asterisks represent the sources under *Sargassum* effect.

| | Mahahual | | | Xahuayxol | | | Xcalak | |
|---|---|---|---|---|---|---|---|---|
| **Genus** | δ¹³C | δ¹⁵N | **Genus** | δ¹³C | δ¹⁵N | **Genus** | δ¹³C | δ¹⁵N |
| *Caulerpa* | −9.89 ± 0.15 | 2.22 ± 0.01 | *Caulerpa* | −8.86 ± 0.19 | 5.68 ± 0.01 | *Caulerpa* | −12.60 ± 0.04 | 1.00 ± 0.10 |
| *Caulerpa** | −16.22 ± 0.55 | 0.93 ± 0.08 | *Dictyota* | −15.71 ± 0.90 | 2.29 ± 0.41 | *Caulerpa** | −9.63 ± 0.02 | 2.02 ± 0.08 |
| *Dictyota* | −16.38 ± 1.23 | 1.56 ± 1.37 | *Dictyota** | −16.31 ± 0.95 | 0.71 ± 0.02 | *Codium* | −12.17 ± 0.07 | 1.25 ± 0.07 |
| *Dictyota** | −15.95 ± 0.04 | 0.82 ± 0.04 | *Halimeda** | −12.61 ± 1.70 | 0.88 ± 0.01 | *Dictyota* | −15.47 ± 0.68 | 0.67 ± 0.03 |
| *Halimeda* | −7.01 ± 1.25 | 0.29 ± 0.43 | *Laurencia* | −14.81 ± 0.23 | 1.36 ± 0.71 | *Dictyota** | −15.69 ± 0.20 | 0.04 ± 0.06 |
| *Halimeda** | −8.39 ± 0.69 | 0.68 ± 0.12 | *Lobophora** | −10.49 ± 1.35 | 0.33 ± 0.64 | *Lobophora* | −14.15 ± 0.53 | 0.77 ± 0.33 |
| *Laurencia* | −16.16 ± 0.90 | 2.61 ± 1.41 | *Penicillus* | −11.51 ± 8.28 | 1.84 ± 0.30 | *Padina* | −10.18 ± 0.18 | 0.25 ± 0.19 |
| *Stypopodium* | −11.33 ± 0.52 | 0.67 ± 0.05 | *Sargassum* | −14.65 ± 1.82 | 3.21 ± 0.23 | *Penicillus* | −14.50 ± 0.08 | 0.06 ± 0.08 |
| Turf | −13.44 ± 0.00 | 3.03 ± 0.02 | *Stypopodium* | −16.80 ± 1.40 | 1.47 ± 0.56 | *Penicillus** | −9.75 ± 0.14 | 1.98 ± 0.04 |
| Turf* | −16.54 ± 0.22 | −0.51 ± 0.02 | Turf | −16.43 ± 1.32 | 1.84 ± 0.30 | *Sargassum** | −14.76 ± 0.87 | 0.37 ± 0.08 |
| *Udotea* | −12.86 ± 0.42 | 2.19 ± 0.03 | Turf* | −18.56 ± 0.04 | −0.89 ± 0.11 | Turf | −17.44 ± 0.48 | 4.59 ± 0.64 |
| | | | *Udotea* | −11.62 ± 1.34 | 2.42 ± 1.12 | Turf* | −21.98 ± 0.10 | 0.41 ± 0.01 |
| | | | *Udotea** | −12.65 ± 0.20 | 2.65 ± 0.77 | | | |
| *S. fluitans* | −16.03 ± 0.99 | −0.53 ± 0.26 | *S. fluitans* | −16.36 ± 0.15 | −1.74 ± 0.38 | *S. fluitans* | −16.26 ± 0.17 | −2.51 ± 0.52 |
| *S. natans* | −17.44 ± 0.71 | −1.59 ± 0.70 | S. natans | −16.82 ± 0.73 | −1.49 ± 0.42 | *S. natans* | −17.28 ± 0.81 | −1.62 ± 0.55 |
| *Sargassum*'s associated turf | −18.29 ± 0.51 | −1.13 ± 0.05 | *Sargassum*'s associated turf | −15.93 ± 0.79 | −0.47 ± 0.07 | *Sargassum*'s associated turf | −16.27 ± 0.63 | −0.96 ± 0.01 |

the lowest mean values of $\delta^{15}$N ($-2.51 \pm 0.52‰$). Turf associated to floating *Sargassum* $\delta^{15}$N values fluctuated from $-0.42‰$ to $-1.17‰$. The most depleted $\delta^{15}$N was exhibited at Mahahual ($-1.13 \pm 0.05‰$) and the less negative mean value was displayed in Xahuayxol ($-0.47 \pm 0.07‰$).

## Sea urchins

There were significant differences $\delta^{15}$N among sites (ANOVA $df = 2$, $F = 6.473$, $p = 0.005$) and the interaction between site*effect (USE and WSE) showed significant differences (ANOVA, $df = 2$, $F = 7.321$, $p = 0.003$).

   *D. antillarum* exhibited no differences among sites for $\delta^{13}$C values $p > 0.05$. However, we found significant differences analysing the USE and WSE effect (ANOVA $df = 1$, $F = 5.301$, $p = 0.03$). The isotopic ratios of *D. antillarum* (USE) varied from $3.83‰$ to $6.13‰$ for $\delta^{15}$N, while $\delta^{13}$C ranged from $-9.41‰$ to $-13.62‰$. Mahahual was the site with the highest average values for $\delta^{15}$N $5.80 \pm 0.30‰$, while Xcalak displayed the lowest average value $4.38 \pm 0.29‰$. The isotopic ratios of *D. antillarum* (WSE) ranged from $4.69‰$ to $6.16$ for $\delta^{15}$N, while $\delta^{13}$C fluctuated from $-8.83‰$ to $-13.42‰$. We found significant differences for $\delta^{15}$N for sea urchins between sites (USE, ANOVA, $df = 2$, $F = 6.47$, $p < 0.005$).-Xcalak showed particularly low values under *Sargassum* effect (average value $4.38 \pm 0.29‰$ *versus* WSE average value $5.44 \pm 0.36‰$). Nevertheless, $\delta^{13}$C exhibited no significant differences although we noticed a negative trend in the values of $\delta^{13}$C under *Sargassum* effect (USE).

## Algal source contributions (SIAR)

Mixing models provided evidence for the contribution of different algal resources for three sites USE and WSE (Table 3). SIAR analysis showed that *D. antillarum* behaved as an opportunistic grazer under the *Sargassum* effect, it is important to note that pelagic *Sargassum,* despite being one of the most abundant available resources, was not the most assimilated resource (Fig. 4). Relatedely, there was a reduction in benthic food sources USE (Fig. 4). Without *Sargassum* effect *D. antillarum* consumed, *Laurencia, Stypopodium* and *Udotea* (12–15% in average) at Mahahual; *Caulerpa, Laurencia, Penicillus, Sargassum* and *Stypopodium* (8–14% in average) at Xahuayxol; and *Codium, Lobophora* and *Padina* (13–15% in average) at Xcalak. Nevertheless, those resources were absent in the diet of *D. antillarum* under *Sargassum* effect (Table 3). Hence, the species displayed differential resource assimilation and *Caulerpa* was the most important resource for *D. antillarum* in Mahahual WSE (up to 37%), followed by Turf (up to 34%) and *Halimeda* and *Udotea* (up to 29% for both). USE the most important resource was *Halimeda* (up to 44%) followed by *Caulerpa* and *Dictyota* (both up to 31% of contribution). *S. fluitans* and *S. natan* s were no important sources (0–28% and 0–23% respectively), and turf associated to *Sargassum* blooms was the lesser assimilated resources by *D. antillarum* from 0 up to 22% (Table 3).

   At Xahuayxol WSE *D. antillarum* showed *Caulerpa* was the most important resource for *D. antillarum* (from 2 up to 25%) and for the rest of algae there were very similar algal contribution (from 0 up to 23%). The main macroalgal contributor of USE was *Udotea* with up to 61%, followed by *Halimeda* and *Lobophora* (with up to 35% and 38% respectively) as secondary resources. *Sargassum's* associated turf showed evidence of low

**Table 3  Average percentage (%) contribution of algal genus to the diet of the sea urchins *D. antillarum* considering the effect of *Sargassum*: without *Sargassum* effect (WSE) and under *Sargassum* effect (USE) at Mahahual, Xahyayxol and Xcalak produced by the SIAR model using isotope values from algae.** Minimum and maximum values for each algae are shown in parentheses.

| Mahahual | | | Xahuayxol | | | Xcalak | | |
|---|---|---|---|---|---|---|---|---|
| **Genus** | **WSE** | **USE** | **Genus** | **WSE** | **USE** | **Genus** | **WSE** | **USE** |
| *Caulerpa* | 19 (1–37) | 14 (0–31) | *Caulerpa* | 14 (2–25 ) | – | *Caulerpa* | 14 (0–29) | 22 (0–40) |
| *Dictyota* | 9 (0–22) | 14 (0–31) | *Dictyota* | 11 (0–22) | 9 (0–23) | *Codium* | 15 (0–30) | – |
| *Halimeda* | 16 (1–29) | 31 (17–44) | *Halimeda* | 10 (0–19) | 17 (0–35) | *Dictyota* | 12 (0–26) | 11 (0–26) |
| *Laurencia* | 12 (0–25) | – | *Laurencia* | 11 (0–21) | – | *Lobophora* | 13 (0–27) | – |
| *Stypopodium* | 12 (0–25) | – | *Lobophora* | – | 20 (0–38) | *Padina* | 15 (0–29) | – |
| Turf | 17 (0–34) | 11(0–26) | *Penicillus* | 8 (0–18) | – | *Penicillus* | 12 (0–26) | 22 (4–39) |
| *Udotea* | 15 (0–29) | – | *Sargassum* | 12 (0–23) | – | Turf | 19 (0–45) | 8 (0–19) |
| *S. fluitans* | – | 12(0–28) | *Stypopodium* | 11 (0–21) | – | *Sargassum* | – | 13 (0–29) |
| *S. natans* | – | 9(0–23) | Turf | 11 (0–22) | 6 (0–16) | *S. fluitans* | – | 7 (0–18) |
| *Sargassum*'s associated turf | – | 9 (0–22) | *Udotea* | 12 (0–23) | 28 (2–61) | *S. natans* | – | 8 (0–19) |
| | | | *S. fluitans* | – | 6 (0–17) | *Sargassum*'s associated turf | – | 9 (0–23) |
| | | | *S. natans* | – | 6 (0–17) | | | |
| | | | *Sargassum*'s associated turf | – | 8 (0–21) | | | |

contribution (from 0 up to 21%) and *S. fluitans, S. natan* s had negligible contribution to *D. antillarum* diet with a maximum of 17% of the proportional contribution (Table 3).

Turf was the main algal resources for *D. antillarum* in Xcalak WSE (up to 45%) followed by *Caulerpa*, *Codium* and *Padina* as secondary resources (close to 30% maximum of contribution); contrasting USE the main macroalgal contributors in Xcalak were *Penicillus* and *Caulerpa* with up to 39% and 40% respectively. Likewise *Dictyota* and *Sargassum polyceratium ( benthic Sargassum)* were secondary resources up to 26% and 29%, respectively. The pelagic components in the other reef lagoons were negligible contributors for *D. antillarum* diet with just 18–23% of maximum contribution (Table 3, Fig. 4).

## Trophic Levels

The overall trophic level data for *D. antillarum* (TL) ranged from 1.97 to 3.22. The species exhibited significant differences among sites (ANOVA $df = 2$, $F = 10.63$, $p = 0.0004$), and exhibited significant differences between WSE and USE (ANOVA, $df = 1$, $F = 17.7$, $p = 0.0003$). Likewise, calculating the interaction between site*effect (USE and WSE) revealed significant differences (ANOVA, $df = 2$, $F = 12.65$, $p = 0.0001$). The highest TL values were reported for Mahahual USE, while the lowest one was recorded in Xahauayxol WSE. At Mahahual, the TL mean value of *D. antillarum* was $2.35 \pm 0.18$ WSE and $3.08 \pm 0.13$ USE; at Xahuayxol, the TL mean value was $2.13 \pm 0.30$ WSE and $2.49 \pm 0.27$ USE, and at Xcalak TL mean value was $2.62 \pm 0.15$ WSE and $2.45 \pm 0.12$ USE (Table 4).
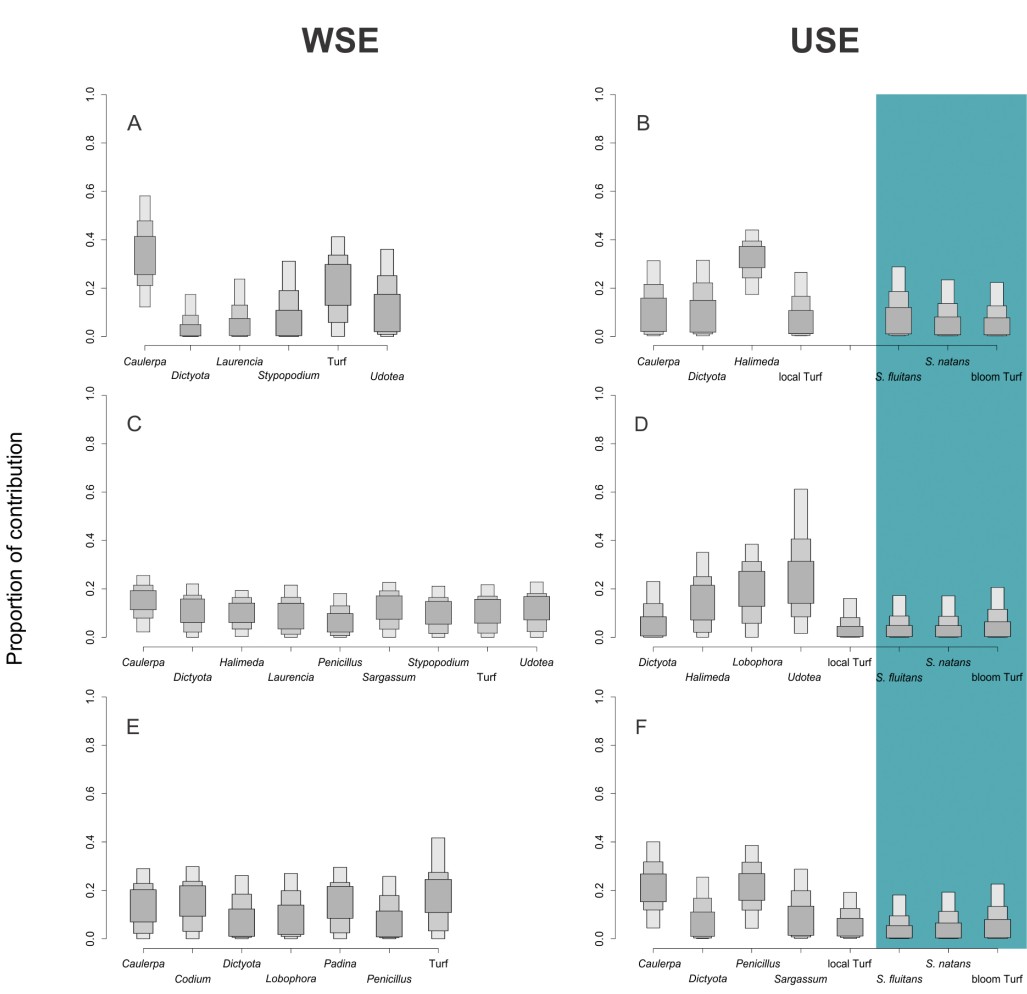

**WSE** **USE**

Proportion of contribution

Source (algae)

**Figure 4 Algal resources proportions consumed by *Diadema antillarum*.** Contribution rates of algae to the diet of *Diadema antillarum* in the two scenarios (WSE and USE). Results are shown as 25% (light error bars), 75% (grey error bars) and 95% (dark error bars) of credibility intervals. (A) Represents the contribution for *D. antillarum* at Mahahual without *Sargassum* effect (WSE), (B) represents *D. antillarum* at Mahahual under *Sargassum* effect (USE); (C) represents *D. antillarum* in Xahuayxol WSE, (D) represents *D. antillarum* in Xahuayxol USE; (E) represents *D. antillarum* in Xcalak WSE and (F) represents *D. antillarum* in Xcalak USE. Bloom turf is the *Sargassum*'s associated turf. The blue bar represents the pelagic sources USE.

**Table 4 Trophic level of *D. antillarum*.** Mean Trophic level (TL), and $\delta^{15}N$ and $\delta^{13}C \pm$ standard deviation of *D. antillarum* without *Sargassum* effect (WSE) and under *Sargassum* effect (USE) at Mahahual, Xahuayxol and Xcalak.

| Site | TL WSE | TL USE | $\delta^{15}N$ WSE | $\delta^{15}N$ USE | $\delta^{13}C$ WSE | $\delta^{13}C$ USE |
|---|---|---|---|---|---|---|
| **Mahahual** | $2.35 \pm 0.18$ | $3.08 \pm 0.13$ | $5.22 \pm 0.43$ | $5.8 \pm 0.3$ | $-10.46 \pm 0.6$ | $-12.32 \pm 0.95$ |
| **Xahuayxol** | $2.13 \pm 0.3$ | $2.49 \pm 0.27$ | $5.09 \pm 0.71$ | $4.9 \pm 0.24$ | $-11.5 \pm 0.81$ | $-11.21 \pm 1.48$ |
| **Xcalak** | $2.62 \pm 0.15$ | $2.45 \pm 0.12$ | $5.44 \pm 0.18$ | $4.38 \pm 0.29$ | $-10.58 \pm 2.01$ | $-12.02 \pm 0.89$ |

**Table 5** **Trophic niche breadth of sea urchins without *Sargassum* effect (WSE) and under *Sargassum* effect (USE) at Mahahual, Xahuayxol and Xcalak calculated by SIBER analysis of muscle values.** SEAc, corrected standard ellipse area.

| Niche breadth | Mahahual | | Xahuayxol | | Xcalak | |
|---|---|---|---|---|---|---|
| | WSE | USE | WSE | USE | WSE | USE |
| SEA | 0.62 | 0.71 | 1.79 | 2.97 | 2.32 | 2.32 |
| SEAc | 0.83 | 0.89 | 2.68 | 3.57 | 3.48 | 0.14 |

## Isotopic Niches

Table 5 shows data on isotopic niche breadth as measured by the corrected standard ellipse area (SEAc). The Stable Isotope Bayesian Ellipses in R (SIBER) analysis suggested a reduction in trophic niche particularly in Xcalak. This site showed the main difference in the trophic niche breadth with SEAc of 3.48 and 0.14 (WSE and USE respectively). An overlap of isotopic niches between WSE and USE was only found in Xahuayxol (Fig. 5). SEAc was higher USE in this site with 3.57 versus 2.68 SEAc WSE (Fig. 5).

## DISCUSSION

Our results provide evidence of the detrimental effect of *Sargassum* blooms on the physicochemical water properties and ecological processes in near-shore coral reef communities as recently has been identified in our study area (*Rodríguez-Martínez, van Tussenbroek & Jordán-Dahlgren, 2016*; *van Tussenbroek et al., 2017*; *Cuevas, Uribe-Martínez & Liceaga-Correa, 2018*). Particularly, the results provide evidence for the input of external carbon and nitrogen resulting from *Sargassum* blooms on benthic communities that alter the nutrient inputs and trophic niche for *D. antillarum*. These findings contribute to the growing recognition of the role of exogenous nutrient enrichment in modifying natural sources in a food web. Hence the organic matter inputs from *Sargassum* coupled with hypoxia leads to modification of natural algal resources for *D. antillarum*. Considering the detrimental effects this likely represents a nutrient limitation to sea urchin herbivory.

Onshore *Sargassum* exhibits physical processes of fragmentation, decomposition and remineralization by bacteria, meiofauna and grazers (*Colombini & Chelazzi, 2003*). The algae-derived organic matter, product of that decomposition, has an effect on *in situ* oxygen availability (*Haas et al., 2010*). *Sargassum* blooms clearly showed a negative impact hypoxic conditions found at the three studied reef lagoons (Fig. 3). This could ultimately drive the success of the communities' nitrogen fixation, evidenced by depleted values of $\delta^{15}$N as reported by *Dorado et al. (2012)* and *France (1995)*.

The dissolved oxygen values in the back reefs of our study areas were lower than the standard values for coral reefs dominated by algae (7.9 $\pm$ 0.5 mg l$^{-1}$) according to *Haas et al. (2010)* and values reported by *Camacho-Cruz et al. (2019)* for Xahuayxol and Mahahual. This supports ideas from *Kendrick et al. (2000)* and *Haas et al. (2010)*, who argue that benthic communities linked to reef lagoons are very susceptible to environmental degradation. Some benthic algae play an important play in the transfer of energy and can be catalyzers of oxygen dynamics in reefs due to coral reef associated algae-derived organic matter (*Wild et al., 2010*).

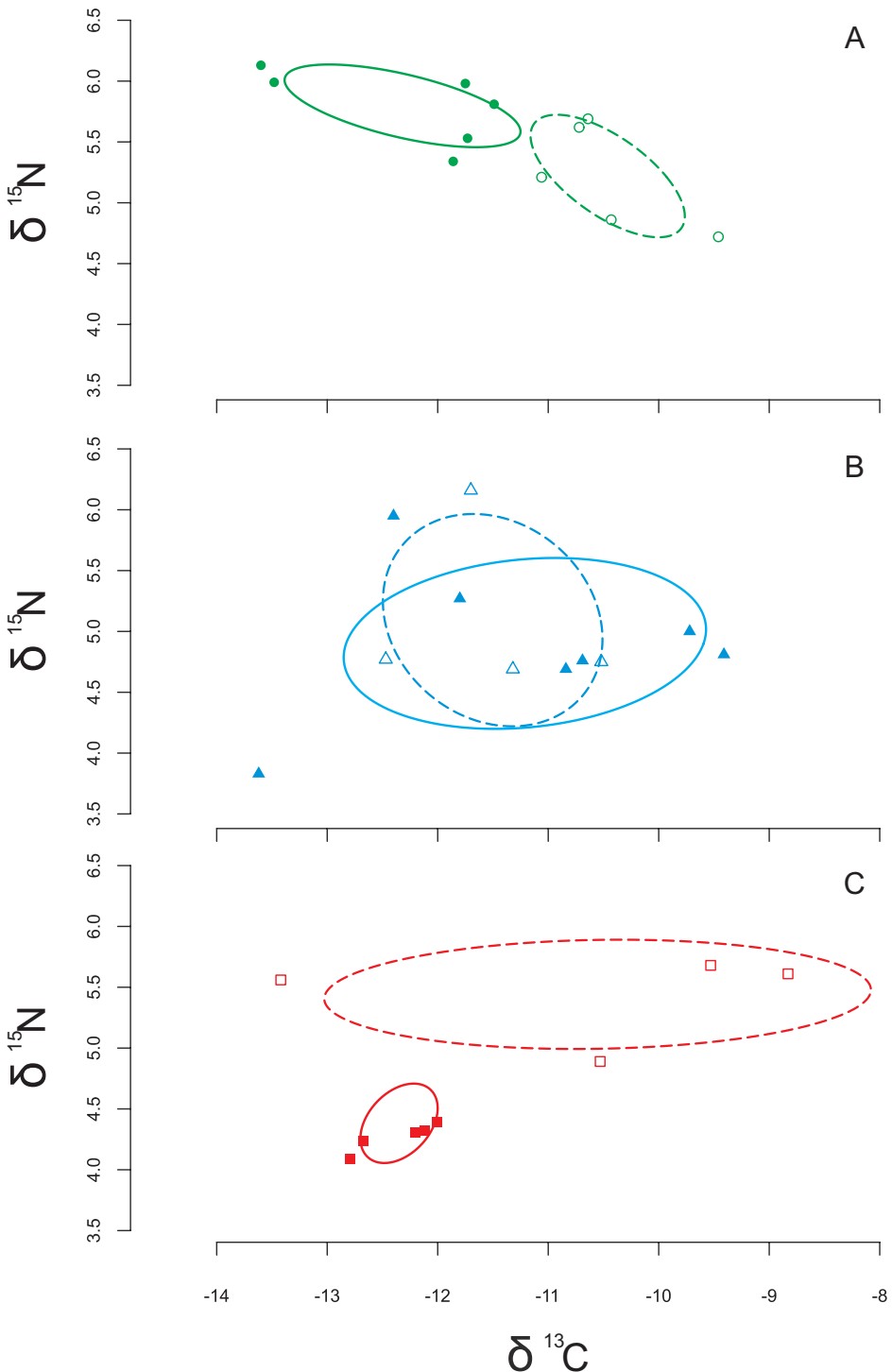

**Figure 5** **Isotope niche breadth of the sea urchin *Diadema antillarum*.** Isotope niche breadth of the sea urchin *Diadema antillarum* at Mahahual (A), Xahuayxol (B) and Xcalak (C). Dotted lines are without *Sargassum* effect (WSE) and solid lines under *Sargassum* effect (USE).

## Isotopic variations in the algal resources

We found that the composition of benthic macroalgae assemblages were different under *Sargassum* and without *Sargassum* effect. USE showed a reduction in the taxonomic diversity of macroalgal food sources available to *D. antillarum* and isotope values presented substantially lower $\delta^{15}N$ values (Table 2). The fact that there were fewer available algal sources in the USE condition implies that the trophic chain becomes less complex as the interaction of primary consumers with their resources is reduced (*Phillips & Gregg, 2003*).

Overall $\delta^{13}C$ values ranged from $-21.98$ to $-5.65‰$ are similar to ranges reported by *Fry & Sherr (1984)* and *Morillo-Velarde, Briones-Fourzán & Álvarez Filip (2018)*. Those authors reviewed the $\delta^{13}C$ data of benthic algae, noting that values ranged between $-30$ and $5‰$. $\delta^{15}N$ overall algae values fluctuated from 0.02 to 5.68‰. Despite these values agree with the variation reported in other studies like *Owens (1987)* and *France (1995)*, we found USE very low, ergo according to *Lapointe et al. (2005)* and *France et al. (1998)*. These low $^{15}N{:}^{14}N$ ratios can be indicative of macroalgae living in oligotrophic reefs which experience nitrogen fixation (*Montoya, Carpenter & Capone, 2002*). In the presence of the leachates of decomposing *Sargassum*, it is possible that anaerobic bacteria gained significance over other benthic groups (Table 2), (*Carpenter & Cox, 1974*; *Rooker, Turner & Holt, 2006*), and could be the cause of the low macroalgal isotopic signatures. On the other hand, high values of $\delta^{15}N$ in macroalgae are linked to land-based N enrichment sources, being a good indicator of anthropogenic nitrogen inputs (*Umezawa et al., 2002*) such as sewage discharges (*Risk et al., 2009*; *Lapointe et al., 2011*).

*France (1995)* reported nitrogen ranges of marine macroalgae from $-3$ to $18‰$. The inconsistencies in this pattern with values of $\delta^{15}N$ close to atmospheric signature of 0% suggest a fixation of nitrogen. *Dorado et al. (2012)* associated the depleted values of $\delta^{15}N$ with nitrogen fixation and its impact on the trophic position of consumers. So, temporal difference between values in this study WSE and USE might be explained by the influence of organic input derived from floating *Sargassum* dragged components. We considered that it is likely that the *Sargassum* effect modifies organic matter dynamics. These modifications stem from changes in the oxygen levels, which were consistently reflected in the low $\delta^{15}N$ values we recorded of for the primary producers.

## Status of *Diadema antillarum* in the Mexican Caribbean

It is important to note that we focused our study on the most abundant species at the three localities and the most important shallow-bottom herbivore on Caribbean reefs (*Carpenter, 1981*; *Hughes, 1994*; *Aronson & Precht, 2006*; *Kissling et al., 2014*). For the Mexican Caribbean, there has been considerable variation in *D. antillarum* population data. *Jordán-Garza et al. (2008)* showed a high presence of *D. antillarum* with densities of more than 7 ind m$^{-2}$ in several areas, including our study area. *Jorgensen, Espinoza-Ávalos & Bahena-Basave (2008)* reported densities of 12.6 ind m$^{-2}$ after hurricane Dean. According to *Maldonado-Sánchez (2018)* population density of *D. antillarum* displayed <1 ind m$^{-2}$ for five different habitats of the Parque Nacional Arrecifes de Xcalak (PNAX) reef lagoon (back reef, seagrasses, sandy bottoms and reef patches) and the fore reef. The back reef exhibited the highest abundance with an average of 0.5 ind m$^{-2}$. However for Mahahual,

we registered an average density of 0.6 ind m$^{-2}$ (N Cabanillas-Terán, pers. obs., 2017), because of the broad variability exhibited in *D. antillarum* populations from the back reef.

## Trophic parameters of *D. antillarum*

Our results support the evidence that *Sargassum* blooms impacted δ$^{15}$N differentially among sites, as the ratios of δ$^{15}$N and δ$^{13}$C are determined by their resources (*Phillips & Gregg, 2003*). It was conspicuous that *D. antillarum* showed higher δ$^{15}$N values USE at Mahahual.

Although some available resources (e.g., *Dictyota* and turf) were present in both conditions (WSE and USE), measuring the contribution of algae to the sea urchin tissues can display key information about how consumers assimilate habitat resources and this could reveal information on the degree of disturbance (*Layman et al., 2007b*). Therefore, it is possible that the ecological role of *D. antillarum* was different in each site and could be explained by the variation in the number of available resources and a differential assimilation (Table 3). The higher δ$^{15}$N values USE in the muscle of *D. antillarum* were a result of the synergistic effect determined by resource availability and disturbance condition.

Pelagic sources may provide new sources of food and the possible nitrogen fixation carried out by turf attached to pelagic *Sargassum* undoubtedly brought a new source of organic matter to basal trophic levels (*Rooker, Turner & Holt, 2006*). However, those sources were not major contributors for *D. antillarum* and appear to avoid the invasive pelagic macroalgae. This is consistent with the feeding ecology by marine generalist herbivores (*Boudouresque & Verlaque, 2001*) and such feeding response is in line with evidence from other sea urchin species in the face of other invasive resources. The experiments carried out by *Tomas, Box & Terrados (2011)* provide evidence that some seaweed invaders were strongly avoided by *Paracentrotus lividus* and therefore escape enemy control by reducing herbivore preference.

The trophic level metric is very useful because the classical discrete trophic level definitions ignore the value of food web connections, omnivory, and diet changes (*Polis & Strong, 1996*; *Vanderklift, Kendrick & Smit, 2006*). Generally the sea urchin *D. antillarum* has been considered as a generalist herbivore (*Ogden & Lobel, 1978*; *Sammarco, 1980*; *Solandt & Campbell, 2001*; *Weil, Torres & Ashton, 2005*). *Morillo-Velarde, Briones-Fourzán & Álvarez Filip (2018)* found that for the North of Quintana Roo Mexico *D. antillarum* occupied an herbivorous trophic position. However, invertebrate samples have been found in the stomach contents this species in the Caribbean, suggesting omnivorous behaviour (*Rotjan & Lewis, 2008*; *Rodríguez-Barreras et al., 2015*; *Rodríguez-Barreras et al., 2016*).

The mean trophic level for *D. antillarum* exhibited at Mahahual was 2.35 ± 0.18 WSE up to 3.08 ± 0.13 USE. Hence, WSE supported the idea that this species occupies an herbivorous position. However USE *D. antillarum* revealed that the species can occupy different trophic niches when faced with resource limitation. Under *Sargassum* blooms, *D. antillarum* displayed a position more in line with omnivorous conditions, suggesting trophic level indicative of herbivorous behaviour tending towards omnivory, according to *Vander Zanden & Rasmussen (1999)*. These authors stated that primary consumers
have a trophic position of 2.0 (strictly herbivorous); but if organisms assimilate primary consumers, they are considered to be a trophic level of 3.0. The results for Mahahual are consistent with *Andrew (1989)* who argued that sea urchins could take advantage of ecosystem changes through omnivory if variation exists in the availability of resources. Our results suggest that *D. antillarum* behave as a facultative omnivore depending on patterns of nutrient availability. $\delta^{15}$N signatures for *D. antillarum* in Mahahual suggest a different carbon source USE. These signatures are also likely the result of anthropogenic nitrogen inputs, as this site has a high eutrophication, being an area with elevated touristic demand (*Martínez-Rendis et al., 2016*; *Arias-González et al., 2017*). Furthermore, possible nitrogen fixation by anaerobic bacteria as an important factor in the variation of available sources of food.

Regarding the TL values exhibited for *D. antillarum* in Mahahual USE 3.08 ± 0.13 versus 2.35 ± 0.18 for WSE would place *D. antillarum* in an omnivorous position tending towards carnivory. Similar values were obtained from Mediterranean sea urchins as a strategy to avoid exclusion by sympatric species (*Wangensteen et al., 2011*). However, we cannot state that *D. antillarum* is carnivorous in Mahahual. This would require a more complete temporal study, and an adjustment of a new $\delta^{15}$N baseline for primary producers, considering that $^{15}$N/$^{14}$N ratios can vary spatially and temporally (*Jennings et al., 1997*; *Vanderklift, Kendrick & Smit, 2006*).

The results for Xahuayxol showed also a trend towards higher $\delta^{15}$N. However by analyzing the condition of *D. antillarum* in Xahuayxol no significant differences were observed. We can assume that this locality was least changed in its foraging behavior position against the nutrients modification and the species occupied a lower trophic level WSE. Meanwhile, Xcalak displayed the opposite trend compared to Mahahual and Xahuayxol and USE *D. antillarum* trophic level was lower than WSE. Our results suggest that for Xcalak the effect of *Sargassum* blooms completely modified and reduced the possibility for finding available resources, displaying a trophic level around 2.5 between the two scenarios of *Sargassum* blooms. This corresponds to a predominantly herbivorous to omnivorous condition. Moreover this was confirmed with the isotopic niche breadth data where a reduced niche was observed for Xcalak (Fig. 3).

The rank found for *D. antillarum* in this study is consistent with the study conducted by *Rodríguez-Barreras et al. (2015)* in Puerto Rico where microinvertebrates were used as source of organic matter by the sea urchin. Finally, TL values support the premise that echinoids are able to modify their foraging behaviour depending on the availability of resources (*Randall, Schroeder & Starck, 1964*; *Muthiga & McClanahan, 2007*), and in this case under *Sargassum* blooms condition was not only determined by macroalgae availability, but for unusual conditions that caused a shift in the ecosystem (*Cabanillas-Terán et al., 2016*).

## Isotopic niche breadth

The ellipses provide integrated information on the relationship between the availability of sources and the niche width. The results of Mahahual indicated that in USE. *D. antillarum* consumes different carbon and nitrogen sources (Fig. 4).

Several studies (*Lawrence, 1975*; *Carpenter, 1981*; *Sammarco, 1982*; *Hay & Fenical, 1988*) noted that echinoids have the ability to adapt their foraging behavior depending on algae availability as well as their population density and site characteristics (*Bak, Carpay & De Ruyter Van Steveninck, 1984*; *Bak, 1994*; *Alvarado et al., 2016*). We observed at Mahahual that USE *D. antillarum* exhibited a broader trophic niche than WSE. Despite the limited resources this could lead to trophic overlap and stronger habitat degradation. SIAR results showed a resource shift and this could be explained in terms of omnivory as stated by *France et al. (1998)* "omnivory is a prevalent attribute of aquatic food webs".

The trophic niche of Xahuayxol reflects that there was no difference in the use of carbon and nitrogen sources. It is noteworthy that for the case of Xcalak, the resulting isotopic niche of *D. antillarum* was significantly smaller under *Sargassum* effect. This is consistent with the metric that associates smaller niche amplitude with disturbed ecosystems (*Layman et al., 2007b*).

## Limitations of the study

To assess the effect of differential management of *Sargassum* and to effectively evaluate the effect of disposal management, quantitative information on beach disposal would be necessary.

From our results, it is clear that algae communities were modified due to *Sargassum*. However, due to the structuring role of sea urchins, and, considering that algae respond to temporal variability naturally, it would be necessary to study changing gradients at different time scales. Such a temporal study would provide more conclusive information about the effect of *Sargassum* spp. on benthic communities.

It is necessary to strengthen the sampling effort to evaluate current population status. A more comprehensive discussion would need to include the interactions with other herbivorous/omnivorous species, that coexist at each site and whether, or how they carry out resource partitioning.

The metrics used in this study allowed us to evaluate the variation of the isotopic signatures that formed the trophic spectrum of *D. antillarum* under two different scenarios. Metric values based on an instantaneous characterization of a single food web provide a limited view of the food web. Therefore, to evaluate the trophic structure and consequently its functional structure, the most promising evaluations would have to include a comparison of multiple gradients, and, to examine the same food web on a longer temporal perspective.

The deposited biomass regarding to *S. fluitans* and *S. natans* did not include a measurement of the total arrived *Sargassum* blooms. However, our results established a baseline for the amounts that were more available for the echinoids that inhabit the back section of the Caribbean shallow reefs.

It would be challenging to evaluate the ecological role of other coexisting species (*Echinometra viridis*, *E. lucunter* and *Eucidaris tribuloides*), and to include samples of micro-invertebrates. However, this could offer new clues to the connectivity between sympatric species, including trophic loops and successional states of algal communities (*Camus, Daroch & Opazo, 2008*) within the benthic communities of coral reefs.

## CONCLUSIONS

The present study provides an initial review of how trophic parameters of *D. antillarum* were modified by the impact of pelagic *Sargassum* blooms in the Mexican Caribbean. The results indicated that the effects of the leachates generated by the decomposition process, the input of organic material and deposition in its vegetal structures modify the organic matter in the environment and hence the isotopic signatures. This has negative consequences in the benthic trophic structure, limiting the natural herbivory of *D. antillarum*. The source of available carbon and nitrogen was modified, and the isotopic signatures of macroalgae associated with the reef sites exhibited significantly lower values of $\delta^{15}$N. Consequently, the trophic niches were changed and in the case of Xcalak, significantly reduced.

## ACKNOWLEDGEMENTS

We acknowledge Katie Cramer, Gerald Islebe and two anonymous reviewers for their valuable and helpful comments on the manuscript. We thank Comisión Nacional de Áreas Naturales Protegidas (CONANP) for logistic support and Alberto de Jesus-Navarrete for lending the multi-parameter water quality checker. The first author is grateful to María Alfaro-Padilla, Roberto Herrera-Pavón and James Boon for their logistic help during fieldwork and laboratory work, to Alejandro Aragón for his help in editing Fig. 1 and helping to modify Fig. 2, and Rebecca Friedel is acknowledged for improving the English of the manuscript.

### Funding

This work was funded by Secretaría de Medio Ambiente y Recursos Naturales (SEMAR-NAT) through the project: OGRMIS- DAC-UCR #001/2015 ECOSUR/SEMARNAT. The funders had no role in study design, data collection and analysis, decision to publish, or preparation of the manuscript.

### Grant Disclosures

The following grant information was disclosed by the authors:
OGRMIS- DAC-UCR: #001/2015.

### Competing Interests

The authors declare there are no competing interests.

### Author Contributions

- Nancy Cabanillas-Terán conceived and designed the experiments, performed the experiments, analyzed the data, contributed reagents/materials/analysis tools, prepared figures and/or tables, authored or reviewed drafts of the paper, approved the final draft.
- Héctor A. Hernández-Arana conceived and designed the experiments, performed the experiments, analyzed the data, contributed reagents/materials/analysis tools, authored or reviewed drafts of the paper, approved the final draft.

- Miguel-Ángel Ruiz-Zárate and Alejandro Vega-Zepeda performed the experiments, analyzed the data, contributed reagents/materials/analysis tools, authored or reviewed drafts of the paper, approved the final draft.
- Alberto Sanchez-Gonzalez analyzed the data, contributed reagents/materials/analysis tools, authored or reviewed drafts of the paper, approved the final draft.

### Field Study Permissions

The following information was supplied relating to field study approvals (i.e., approving body and any reference numbers):

The collection permit (PPF/DGOPA-002/17) was obtained from the Comisión Nacional de Acuacultura y Pesca (CONAPESCA).

### Data Availability

The raw data are available in the Supplemental Files.

### Supplemental Information

Supplemental information for this article can be found online at http://dx.doi.org/10.7717/peerj.7589#supplemental-information.

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
