# Peer review of "Sargassum blooms in the Caribbean alter the trophic structure of the sea urchin Diadema antillarum"

_PeerJ, doi:10.7717/peerj.7589_

## Round 0.1 · original submission · Major Revisions

Dear Nancy and co-authors,
I just received all the reviews of your manuscript. Although all reviewers consider the study very interesting and providing new findings on the topic, some issues need to be considered before the acceptance. Please, consider all comments and suggestions but pay special attention to those provide by the reviewer#1. All reviewers have included annotated PDFs. Don't forget to include a letter response along with the revised version of the manuscript. In this letter you must respond point by point to each question.

·

Basic reporting

I appreciate how difficult it can be to write a technical paper in another language, and the authors have made a good effort. However, grammar and sentence structure need to be much improved throughout the paper. Many sentences are too long and could be split into two (or three) sentences. Other sentences are fragments.

Additional major improvements (in order of importance) include:
1. Consider possibly narrowing the focus of the study. The study measures numerous values that make it hard to remember the overall aims of the study. Is it to determine how Diadema’s diet changed after the influx of Sargassum on the coast? Is it to quantify the trophic or niche breadth of Diadema? Is it to measure how Sargassum decomposition resulted in hypoxia in lagoonal backreef environments? Each of these questions are valid, but together, the various measurements reported in the study do not tell a cohesive story.
2. Clearly state a hypothesis/hypotheses in the Abstract and Intro. Currently, the wording is very vague and no specific predictions are made about the impact of Sargassum on Diadema. This leaves the reader with many unanswered questions: (a) Why did the study focus on Diadema? It was a keystone herbivore on Caribbean reefs whose mass dieoff seems to have precipitated a coral/algal phase shift - is this why? (b) how would Sargassum be expected to impact Diadema’s diet? Would it limit benthic macroalgal resources by shading? No mechanisms are provided for how Sargassum would be predicted to impact this urchin.
3. The Introduction needs to be re-written so that it clearly and succinctly sets up the study. The intro should explain the (a) overarching aims of the study (there seem to be too many – see comment #1), (b) how each of the stated study questions were investigated (currently, the rationale for collecting each specific measurement is not clearly stated), (c) how the various measurements can be integrated or considered together to arrive at a specific conclusion. The order of the paragraphs needs to be rearranged so that the rationale for using a particular technique (for example, stable isotopes) is clearly explained before going into the technical details of the technique. The aims are also stated at the end of the Intro, and should be moved up towards beginning of Intro.
4. The paper says it is measuring how Sargassum has impacted the “natural” trophic structure of Caribbean reefs. But modern reefs have been significantly altered by human and (maybe) natural disturbances. Diadema populations are currently a tiny fraction of their pre-dieoff abundances. This needs to be acknowledged and discussed in the Intro and Discussion.

Experimental design

Significant efforts need to be made to clarify the specific research questions being asked, and why (currently, it appears to be a general “how did Sargassum affect numerous measures of Diadema’s diet and trophic position”, without any context for why this is relevant and meaningful). In addition, the statistical analyses should be modified to be more informative. Specific suggestions include:

1. The Methods section needs to be re-worded to more succinctly provide the details of the study design and laboratory and statistical analyses. The first paragraph in Methods should be moved (the first sentence should go in Study Sites section, and the rest should go in Discussion). The Study Sites section should give more detail about the relative level and type of human disturbance at each site (and this should also be indicated in the table within Figure 1. Figure 1 should also have the boundaries of the Marine Protected Area). Collecting and Processing Data section should have more detail about: (a) choice of July/Aug 2016 for period without Sargassum effect (WSU) – were all Sargassum gone by then? What happened to it – did it get swept offshore, decompose, was removed by managers? Was water clarity back to pre-Sargassum levels?, (b) how were algae sampled – ALL taxa that were collected within a quadrat, or only fleshy, turf, and erect calcareous (were CCAs excluded?) Did a taxon have to be present in a certain abundance to be included in analyses? Were algae collected by scraping (and were any corals harmed in this process)? Why were Sargassum and turf on Sargassum included – how could a benthic urchin eat pelagic macroalgae?, (c) a figure needs to be added to show the study design for collection of algae and urchins. The Data Analyses section needs to include a figure or table showing how isotopic results from the algae were translated into relative contributions of each taxon to Diadema diet. Currently, this is a bit of a black box. How certain were these results?
2. It is stated that no species that were collected are threatened, but Diadema is severely depleted across the Caribbean… was it ecologically damaging to collect these urchins? Or is Diadema is making a comeback on these reefs? Please provide information on the status of this urchin in Quintana Roo.
3. Much more information is needed in Data Analysis section to be able to interpret the results/replicate the study. Rationale needs to be provided for choosing certain fixed values for parameters within the various models. For example, how were the “isotopic discrimination factors” and “theoretical geometric assumptions” chosen for this study determined to be valid for this data? (And what is the definition of these factors?) Why were certain macroalgal taxa/functional groups pooled for the mixing models analyses? For the SIBER analyses, what are the “metrics” that this analysis performs? And what is a “burn in”? What were the results of the tests for homogeneity and normality?
4. The statistical analyses need to be modified to explain the differences among sites. The Methods states that the sites differ by (a) the amount of Sargassum accumulated, (b) the management strategies for removing Sargassum, and (c) the width of the lagoon. The Results show significant differences among sites. Therefore, analyses should include those characteristics of each site (a, b, and c above) as predictor variables of changes in algal community composition or changes in niche breadth before and after Sargassum.

Validity of the findings

Because there are so many different types of analyses that are performed and the results are reported in a somewhat disorganized manner, it is difficult to interpret the results and draw a cohesive message from them. This is compounded by the fact that mechanistic explanations for results are very rarely given. Lastly, confusing or contradictory results are often not properly interpreted in the conclusions, or conclusions are not supported by the results. Some suggestions for improvement:

1. Consider narrowing the set of research questions/analyses. For example, why does algal composition of Diadema diet, Diadema trophic level, AND Diadema niche breadth all need to be measured? The authors do not consider what all three of these measures together might tell us about Sargassum impact on Diadema diet.
2. Consider re-framing the research questions. This study provides a good description of Diadema trophic characteristics, and it also provides valuable data on oxygen limitation due to Sargassum decomposition. Either of these findings could make up its own paper.
3. It is difficult to be certain that changes in each of these measures is directly caused by Sargassum, because the results differ by site. There could also be many other factors at play, such as differences in SST, precipitation, land-based runoff, etc. between the two years. These factors need to be included in analyses of change in Diadema trophic position and diet.
4. The authors appear intent on showing the negative impacts of Sargassum on Diadema, but the results don’t seem to support this conclusion. Benthic macroalgae was higher when Sargassum was present – wouldn’t this be a good thing for Diadema? At 2 out of 3 sites, Diadema trophic level increases with Sargassum presence. Could this mean that Sargassum results in higher quality food for Diadema? Or did Sargassum presence somehow reduce the abundance of Diadema competitors? For one site, Diadema niche breadth increases under Sargassum presence. Overall, the results seem to indicate that Sargassum either has a neutral or positive effect on Diadema (although these differences cannot be conclusively attributed to Sargassum – see above). Wouldn’t a more straightforward measure of the effect of Sargassum on Diadema be to measure Diadema abundance or body weight before and after the presence of this algae?

Reviewer 2 ·

Basic reporting

The paper has a good structure, in fluent (except the discussion, which is a little bit confusing) and sound. The English can be improved, basically shortening some sentences and making clearer some concepts. The references cited are appropriate for the approach made by the authors, and the main hypothesis is correct, from my point of view.

Experimental design

The experimental design has some problems highlighted in the attached document.

Validity of the findings

In general, the findings are novel and deserve publication. The methods used and the stable isotope approach is correct, and the results found are in line with previous work done in other places under other or similar conditions. The authors highlight the limits of their own study, which is a very good approach. Statistics are in line with what we expect from this kind of treatment, but there are some particular points that could improve the approach made by the authors (see attached document).

Additional comments

The paper has a good novel approach of how the Sargassum blooms are impacting coral reef ecosystems and how. In fact, such recurrent biomass degradation in the Caribbean coasts has to have an impact on the trophic relationships in the already stressed coral reef ecosystems. I made some questions and recommendations in the manuscript to be considered by the authors that may help to clarify to the future readers the comprehension of your findings.

Reviewer 3 ·

Basic reporting

The English language is clear and I did not identify any major problems. I would be careful using the term “shoals”. "Shoals" does not seem appropriate for the pelagic Sargassum (line 18). I assume that shoal can be used to things you can count individually, as fishes and crabs. I suggest you to use “mats” instead.

In the line 52, the reference Sissini et al. 2017 is about extensive masses of Sargassum in 2015. Check the others references. You could replace "in 2011" for "Between 2011 and 201x".

Figure 4 – Add the local in the figure instead of the caption.

Caption Figure 5 – I suggest giving more information about the data. For example, specifying which analysis was made.

Experimental design

Lines 368-369 “We found that the composition of benthic macroalgae assemblages were different among both conditions (WSE and USE).” How was the comparison of benthic macroalgae assemblages between years if the quadrats were not fixed? Nine quadrats is enough to characterize the assemblage?

Validity of the findings

This paper shows the physical and ecological effects of the pelagic Sargassum in near-shore coral reef communities. How the benthic macroalgal community responds to the increase of carbon and nutrients and the nutrient flux in the food web under and without the Sargassum effect. It is well delineated and the approach to trophic structure give us a complementar understanding about the impact of pelagic Sargassum on coastal environments.

---

## Round 0.2 · Minor Revisions

Firstly, I appreciate the deeply revision of your ms. I think the work improve a lot but some inconsistencies and weaknesses need to revised before the acceptance. Reviewer#1 Katie Cramer kindly reviewed the revised version of the ms and sent comments. I think she detects some fragments which need to be rewritten in order to improve the quality of the study. Please, consider all comments and review you study incorporating all suggestion provided by Dr. Cramer.

Again, I appreciate your effort and I strongly encourage you to review the ms.

Best regards,

Salva

·

Basic reporting

The authors have made significant improvements to better clarify the aims of the paper and the methods they employed. I appreciate that in their rebuttal letter, the authors answered several of my questions about ecological inferences that can be made from their reported results. But they now need to add these explanations in their manuscript as well! In conclusion, two areas still need improvement before the paper is ready for publication:

(1) Wording is still unnecessarily vague throughout the manuscript, and much clarification is still needed (for example, “trophic dynamics”, “negative effects”, and similarly vague phrases need to be replaced by statements that describe the specific effects they are trying to measure and the specific ecological mechanisms to be inferred from their observed results). This vague wording and lack of mechanistic explanations makes it difficult to draw coherent links between the specific analyses conducted, the results, and conclusions that can be drawn from these results. In the attached annotated manuscript, I have pointed out areas where vague wording needs to be replaced by specific wording that explicitly states what ecological information will be provided by an analysis and what a result means ecologically.

(2) Grammar, sentence structure, and flow of the manuscript still need to be improved to provide sufficient clarity. There are many instances of redundant statements, numerous statements that could be simplified/made more succinct with fewer words/sentences, and many run-on sentences that need to be shortened. In the attached annotated manuscript, I have pointed out specific suggestions for improvement.

Experimental design

The experimental design appears solid. However, as noted above, more justification is needed for the specific sampling design chosen (why these three reefs? why sample each reef at different distances from shore? why take dissolved oxygen measurements?). None of these are explained in the Intro or Methods - please add justification for each aspect of the study.

Validity of the findings

I honestly am still finding it difficult to assess the validity of the main conclusion that Sargassum was found to be detrimental to Diadema. Is this because the taxonomic richness of the algae found in sampled Diadema declined after Sargassum influx? Is this unequivocally a negative for Diadema - does this necessarily mean that Sargassum "limits available resources" for Diadema as stated in the manuscript? What if Sargassum leachates and decomposition increased some of the preferred algal food sources for Diadema and limited some less preferred algae?

I am still not convinced that this study conclusively demonstrates a negative effect of Sargassum on Diadema. But I may be mistaken in this conclusion - it is hard to know, because the specific inferences and conclusions being made from each result are not clearly stated in the Results and Discussion. The above suggestions to add more explanation of inferences from results might help clear this up. Another possibility is that the results are ambiguous with respect to impacts on Diadema. If this is the case, the narrative needs to be modified to "Sargassum caused changes in diet/trophic characteristics of Diadema and caused reef hypoxia, the latter which could be bad for overall coral reef ecosystem functioning".

Please see the attached annotated manuscript for some suggestions for clarifying study results and conclusions.

Additional comments

Thank you for your revisions. This study is definitely interesting and worthwhile from a conservation and scientific standpoint. If a major effort can be made to: (1) use more precise wording, (2) add statements providing why each analysis was undertaken - exactly what information does each analysis provide, (3) add explanation of the precise ecological mechanisms that could explain your results, and (4) proofread for correct grammar and sentence structure, your manuscript will be ready for publication.

---

## Round 0.3 · accepted · Accept

Dear Dr. Cabanillas-Terán,

It is my pleasure to inform you that your manuscript has been officially accepted for publication in PeerJ.

I hope you will be pleased with the final article, and that you will want to submit other manuscripts to PeerJ.

Congratulations!
Salva